# Ileal Bile Acid Transporter Blockers for Cholestatic Liver Disease in Pediatric Patients with Alagille Syndrome: A Systematic Review and Meta-Analysis

**DOI:** 10.3390/jcm11247526

**Published:** 2022-12-19

**Authors:** Hafiza Sidra tul Muntaha, Mubashar Munir, Syeda Haleema Sajid, Zouina Sarfraz, Azza Sarfraz, Karla Robles-Velasco, Muzna Sarfraz, Miguel Felix, Ivan Cherrez-Ojeda

**Affiliations:** 1Department of Research, Services Institute of Medical Sciences, Lahore 54000, Pakistan; 2Department of Research, King Edward Medical University, Lahore 54000, Pakistan; 3Department of Research and Publications, Fatima Jinnah Medical University, Lahore 54000, Pakistan; 4Department of Pediatrics and Child Health, The Aga Khan University, Karachi 74800, Pakistan; 5Department of Allergy and Pulmonology, Universidad Espíritu Santo, Samborondón 092301, Ecuador; 6Department of Medicine, New York City Health + Hospitals, Lincoln, NY 10034, USA

**Keywords:** Alagille syndrome, chronic cholestasis, refractory pruritus, serum bile acids, maralixibat, odevixibat

## Abstract

Alagille syndrome (ALGS) is a rare, debilitating inheritable disease that is associated with refractory pruritus due to chronic cholestasis. The following systemic review and meta-analysis presents the latest evidence for ileal bile acid transport (IBAT) blockers in AGLS patients in order to improve their efficacy. This study adhered to PRISMA 2020 Statement guidelines. A systematic search of PubMed/MEDLINE, Web of Science, Scopus, and the Cochrane library was conducted from inception until 23 October 2022. A combination of the following keywords was used: Alagille syndrome, therapeutics, treatment, therapy. Meta-analytical outcomes included effect directions of end-line changes in serum bile acids (sBAs), Itch Scale scores (ItchRO), Multidimensional Fatigue Scale scores, pediatric quality of life (QL), alanine aminotransferase (ALT), and total bilirubin. A total of 94 patients across four trials were enrolled and received maralixibat, odevixibat, or a placebo. There was a significant reduction in ItchRO scores by 1.8 points, as well as in sBAs by 75.8 μmol/L. Both the Multidimensional Fatigue Scale and Pediatric QL scale were also improved by 11.4 and 8.3 points, respectively. However, ALT levels were raised by 40 U/L. The efficacy of IBAT inhibitors across current trials was noted. Future trials may focus on the optimization of dosing regimens, considering gastrointestinal side effects and drug-induced ALT elevation in AGLS patients.

## 1. Introduction

Alagille syndrome (ALGS), also known as arteriohepatic dysplasia, is a multisystem disease with chronic cholestasis as a major clinical manifestation. Other cardinal features include stenosis in the pulmonary artery, butterfly vertebrae, characteristic facies, posterior embryotoxon, renal dysplasia, and growth failure [1]. It occurs due to a mutation in the Notch signaling pathway, which is *JAG1* (ALGS type 1), but a small number of patients have mutations in *NOTCH2* (AGLS type 2) [2]. The inheritance is typically autosomal dominant but there may be variable expression, and somatic mosaicism has also been observed; as such, other family members may frequently carry the same genetic defect but may manifest only some or none of the features [3]. One of the most distressing symptoms is associated with cholestasis–pruritus [4]. Among patients with severe cholestasis at early age, about 20–70% require liver transplantation before adulthood [5,6,7]. End-stage liver disease as a result of biliary atresia (BA) is a cause for liver transplantation. A population-based study found that an estimated 7% of the general population may have chronic pruritus [8]; while the global prevalence is unknown, a wide array of patients undergo transplantation due to chronic pruritus. However, a study has reported that an estimated 15% of individuals with chronic pruritus require liver transplantation [9,10].

Many medications are used for chronic cholestasis, including ursodeoxycholic acid (UDCA), cholestyramine, rifampin, and naltrexone, but they produce no major improvement in pruritus. Despite having received commonly prescribed medications for pruritus, it was found that there was only partial control of pruritus in 41.2% of patients (*n* = 50), which affected quality of life in 19.6% of them [11]. Another study reported suboptimal response with UDCA, with pruritus resolving in 26.7% of patients (*n* = 15) [12]. Rifampicin also did not improve refractory pruritus, with complete relief only noted in 15% of patients (*n* = 33) and no significant difference in laboratory parameters of patients with AGLS [13]. Although sertraline, a selective serotonin reuptake inhibitor (SSRI), has been effective as an additional therapy for refractory pruritus, its mechanism of action is unclear in pediatric AGLS [14]. Chronic cholestasis is associated with intractable pruritus and poor quality of life [15]. The following systematic review and meta-analysis collates emerging therapies for the medical management of chronic cholestasis in patients with AGLS.

## 2. Materials and Methods

### 2.1. Search Strategy

This study adhered to the Preferred Reporting Items for Systematic Reviews and Meta-Analyses (PRISMA) Statement 2020 guidelines [16]. PubMed/MEDLINE, Scopus, Web of Science, and the Cochrane Library were systematically searched from inception until 23 October 2022. No language restrictions were applied; non-English language studies were translated into English using Google Translate. A combination of the following keywords was used across the databases with Boolean (and/or) logic: Alagille syndrome or (Alagille and syndrome) and therapeutics or treatments or therapy. The titles and abstracts of all studies were screened independently by two reviewers. In accordance with the umbrella methodology, the reference lists of all screened studies were additionally reviewed to ensure no data were missed. A third reviewer (I.C.-O.) was present to resolve any disagreements during the screening phase.

### 2.2. Study Selection Criteria

The inclusion criteria were studies focusing on clinical trials employing pediatric patients aged 18 years or below of any gender with Alagille syndrome who underwent interventions with ileal bile acid transporter blockers for cholestatic liver disease.

Cohorts (retrospective or prospective), case series, case reports, systematic reviews, meta-analytical studies, brief reports, and letters to editors were omitted.

### 2.3. Data Extraction

Two reviewers extracted the data together from the shortlisted trials into a shared spreadsheet. The third reviewer oversaw the entries and was present for any disagreements. The reviewers identified the trials and therapeutics used in the shortlisted studies. The data were extracted into the following domains: title, author, journal, year, phase, design, inclusion criteria, pharmacologic agent and mode of administration, intervention, outcome measures, follow-up, total participants (*n*), age (in years), gender (percentage males), efficacy, safety, and remarks.

For the meta-analysis, data were collected for the following, including change at end-line compared to baseline, standard deviation (SD), and sample size (*n*): serum bile acid, Itch Scale score, Multidimensional Fatigue Scale score, pediatric quality of life, ALT, and total bilirubin.

The study data were collated into a presentable format during the inclusion phase. To omit duplicates, EndNote X9 (Clarivate, London, UK) was the preferred software used during the study selection process. Bibliographic management was conducted using Mendeley (Elsevier, Amsterdam, the Netherlands).

### 2.4. Statistical Analysis

Data were collected for serum bile acid, Itch Scale (ItchRO (Obs)) score, Multidimensional Fatigue Scale score, pediatric quality of life (QL), alanine transaminase (ALT), and total bilirubin. The mean score changes at the end-line compared to the baseline, along with the standard deviation, were entered into the statistical software. The weighted effect direction was computed with the mean difference reported for all outcomes, applying 95% confidence intervals (CI). The *p*-value, τ^2^, and I^2^ value were computed to assess for heterogeneity. The findings were considered statistically significant if the *p*-value was less than 0.05. The findings are presented in the form of forest plots. A minimum of two studies were required to generate a forest plot for a specific outcome. The statistical testing for the meta-analysis was conducted in R (v. 4.2.2). Moreover, Cohen’s coefficient for the inter-reviewer agreement was calculated in the Statistical Package for Social Sciences (IBM® SPSS®, v25, United Kingdom).

### 2.5. Quality Assessment

The included clinical trials were assessed for quality using the GRADE approach as recommended by Cochrane Training, which was used to assess the overall quality of evidence. The factors assessed for quality of evidence included risk of bias, study design, inconsistent results, lack of generalizability, and inaccurate data. The quality of evidence was graded and presented with overall scores of 1—high quality, 2—moderate quality, 3—low quality, 4—very low quality, and 5—no evidence. The findings were tabulated. The GRADE assessment form was assessed and shared with all reviewers and the final scores were agreed upon before finalization.

Two tools were used for the risk-of-bias assessment. The first was Version 2 of the Cochrane Risk-of-Bias Tool for Randomized Trials (RoB 2) [17]. This consists of five domains: (1) bias arising from the randomization process, (2) bias arising due to deviations from the intended interventions, (3) bias arising due to missing outcome data, (4) bias arising in the measurement of the outcome, and (5) bias arising in the selection of the reported result. Author-led judgments were made classifying the RCTs as follows: (1) low risk of bias, (2) some concerns, and (3) high risk of bias.

The second tool was the Risk of Bias in Non-randomized Studies of Interventions (ROBINS-I) tool [18], which consists of seven domains: (1) bias arising due to confounding, (2) bias arising due to the selection of participants, (3) bias arising in the classification of interventions, (4) bias arising due to deviations from the intended interventions, (5) bias arising due to missing data, (6) bias arising in the measurement of outcomes, and (7) bias arising in the selection of the reported result. Reviewer-led judgments were made classifying the non-randomized clinical trials as: (1) low risk, (2) moderate risk, and (3) serious risk. For both tools, a traffic light plot of bias assessment was generated.

## 3. Results

During the identification phase (phase 1), a total of 1621 studies were identified, from which 329 duplicates were removed before the screening. In the screening phase (phase 2), a total of 1292 titles and abstracts were screened. Of these, 1245 studies were excluded before screening the full texts as the titles and abstracts did not warrant inclusion. Finally, 47 full-text studies were assessed for eligibility. Of these, 43 were excluded (Figure 1). Finally, in the inclusion phase (3), four studies were included that reported efficacy and safety outcomes for the pharmacological agents being reviewed in patients with Alagille syndrome. Cohen’s coefficient of the inter-reviewer agreement was computed to be 0.88, suggesting excellent agreement.

### 3.1. Design and Inclusion Criteria

Gonzales et al. [19] recruited 31 pediatric patients with a mean age 5.4 years (standard deviation, SD = 4.25; range = 1–18) with Alagille syndrome who were treated with maralixibat in a placebo-controlled, randomized, phase 2 trial with a parallel-group randomized withdrawal period (RWD). Patients were recruited who had one or more of the following: total sBA > 3 times the upper limit of normal (ULN) for age, conjugated bilirubin > 1 mg/dL, unexplained fat-soluble vitamin deficiency, GGT > 3 times ULN, and/or intractable pruritus explained only by liver disease.

Baumann et al. [20] conducted an open-label, non-randomized, phase 2 trial with different odevixibat dosing regimens. A total of 24 pediatric patients with a mean age of 6.5 years (SD = 4.6; range = 1–18) with pruritus due to chronic cholestatic disease, including 6 patients with Alagille syndrome, were enrolled. Other inclusion criteria were elevated total sBA ≥ 2 times the upper limit of normal (ULN) and a score of ≥3 on an 11 point visual analog scale (VAS) for itch averaged over 7 days.

Shneider et al. [21] reported outcomes of two phase 2, placebo-controlled, double-blind, randomized clinical trials. A total of 57 patients were included with a mean age of 6.5 years and with severe cholestasis (evidence of cholestasis, intractable pruritus, and compensated liver disease) secondary to Alagille syndrome. Patients were enrolled if they had a mean daily ItchRo (Obs) score of ≥2 for two consecutive weeks before enrollment (Table 1).

### 3.2. Dosing Regimens

Gonzales et al. [19] administered maralixibat orally once a day at a dose of up to 380 μg/kg until week 18, after which the RWD was tested. Following 1:1 randomization, half the patients continued receiving maralixibat from weeks 19 to 22 and the other half of the patients received a placebo in the same period. Maralixibat was continued until week 48, followed by a long-term extension of up to 204 weeks and doses increased up to 380 μg/kg twice per day. Baumann et al. [20] tested five doses (10, 30, 60, 100, or 200 μg/kg) for orally administered odevixibat. At first, a single dose was given to patients and a 14 day safety period was observed. The same dose was then given daily for 4 weeks. Shneider et al. [21] provided doses of orally administered maralixibat ranging from 140 to 560 μg/kg from baseline to 48 weeks. Dose escalation and optimization were completed in week 12. The most commonly administered dose was 280 μg/kg daily, but higher and lower doses were also given (Table 2).

### 3.3. Outcome Measures

Gonzales et al. [19] measured the change from weeks 18 to 22 (the RWD period) in fasting sBA among any participant who had a reduction in sBA ≥50% from baseline to week 18 as the primary outcome measure. Secondary outcome measures were changes in sBA from baseline to week 18, ItchRO scores from baseline to week 18 and from week 18 to week 22, and liver enzymes from baseline to week 18 and week 18 to 22 (alkaline phosphatase (ALP), alanine aminotransferase (ALA), total bilirubin (TB), and direct bilirubin (DB)). Other outcome measures reported by Kamath et al. [22] for the same trial were changes from baseline to week 48 in the Itch-Reported Outcome score, Multidimensional Fatigue Scale score, and Pediatric Quality of Life Inventory Generic Core score.

Baumann et al. [20] outlined one primary efficacy endpoint and three secondary endpoints. The primary endpoint was a change in serum bile acid levels from baseline to the end of the 4 week treatment period. The secondary efficacy endpoints were changes in VAS-itch scores [23], Whitington itch scores [24], and Partial Patient-Oriented Scoring Atopic Dermatitis (PO-SCORAD) itch and sleep disturbance scores [25].

Shneider et al. [21] reported the mean change in sBA from baseline to weeks 48 and 72; ItchRo (observer) score (0–4-point scale) from baseline to weeks 48 and 72; Clinician Scratch Score (0–4-point scale) from baseline to weeks 48 and 72; and quality of life measures, including Parent PedsQL and Multidimensional Fatigue Scale (MFS) module (0–100 point scale), from baseline to weeks 48 and 72 (Table 2).

### 3.4. Efficacy

Gonzales et al. [19] found that, during the RWD period between weeks 19 and 22, the placebo group showed significant increases in sBA by 94 μmol/L (95% CI: 23 to 164) and in pruritus by 1.7 points (95% CI 1.2 to 2.2). Among patients in either the placebo or intervention arms who had an sBA ≥50% from baseline to week 18, there was a mean difference in sBA of 117 μmol/L (95% CI: 232 to −2) in the RWD period. Another study by Kamath et al. [22] reported quality of life outcomes within the same trial. A total of 20 out of 27 patients (74%) had an improvement in treatment response of ≥1 point on the Itch-Reported Outcome scale. The mean change in the Multidimensional Fatigue Scale score was significantly higher: +25.8 points in responders versus −3.1 in non-responders. Patients who continued receiving oral maralixibat maintained improvement until week 204.

Baumann et al. [20] reported the mean sBA score from across the entire cohort at the end of 4 week therapy with odevixibat to be −123 ± 118 μmol/L (−394–15), and there was a reduction in the mean sBA in five of six patients with Alagille syndrome. Secondary measures included a VAS-itch score reduction of −6.1 to 0.4 points, a PO-SCORAD itch score reduction of −6.7 to −0.03 points, a Whitington itch reduction of −1.6 to 0.8 points, and a PO-SCORAD sleep change of −5.5 to 0.7 points. There was an improvement in sBA, pruritus, and sleep disturbances with oral odevixibat among children with Alagille syndrome.

Shneider et al. [21] reported significant reductions in the observer-reported ItchRO score of −1.59 (−1.81, −1.36) and the Clinician Scratch Score of −1.36 (−1.67, −1.05) from baseline to week 48. The sBA was also significantly reduced from baseline to week 48 at −78.88 μmol/L (−114.57, −45.19). The Parent PedsQL scale showed a significant improvement of +10.17 points (4.48, 15.86). The Multidimensional Fatigue Scale score also significantly improved by 13.97 points (7.85, 20.08). Similar trends were noted in long-term administration at week 72 (Table 3). 

### 3.5. Safety and Tolerability

Gonzales et al. [19] found predominantly gastrointestinal symptoms with oral maralixibat. Treatment of Alagille syndrome with maralixibat only led to self-limiting mild-to-moderate adverse events. Baumann et al. [20] found two patients (33.33%) with Alagille syndrome who received 200 μg/kg of oral odevixibat had elevated ALT/AST levels that were present at baseline and persisted until the end of 4 weeks. Variations in liver enzyme levels were, however, not directly attributed to treatment, and there were no consistent patterns. Shneider et al. [21] reported a 15.8% incidence in treatment-emergent adverse events (9 of 57 patients); of these, six of the patients showed a rise in alanine aminotransferase (ALT) or total bilirubin (TB) (Table 3).

### 3.6. Meta-Analytical Findings

#### 3.6.1. Serum Bile Acid

All four studies reported serum bile acid findings at the end-line compared to the baseline (Figure 2). The mean difference was MD = −75.804 (95% CI = −104.726, −46.881). The intervention showed favorable mean reductions in serum bile acid among participants. The findings were significant, with no signs of heterogeneity (*p* < 0.0001, τ^2^ = 0, I^2^ = 0%).

#### 3.6.2. Itch Scale: ItchRO (Obs)

All four studies reported outcomes for the Itch Scale at the end-line (Figure 3). The mean difference was in favor of intervention, with the following values yielded: MD = −1.873 (95% CI = −2.161, −1.585, *p* < 0.0001). Overall, there was no heterogeneity (τ^2^ = 0, I^2^ = 0%), and the findings were significant.

#### 3.6.3. Multidimensional Fatigue Scale

Two of the four studies reported findings for the Multidimensional Fatigue Scale (Figure 4). It was ascertained that the mean difference suggested an increase in the overall score on the scale, meaning that the intervention favored fatigue reduction (MD = 11.41, 95% CI = 5.43, 17.38). No heterogeneity was observed, and the results held significance (*p* = 0.0002, τ^2^ = 0, I^2^ = 0%).

#### 3.6.4. Pediatric QL

With regard to the findings of the quality-of-life scale assessment, two of the four studies reported outcomes (Figure 5). The overall mean difference was quantified as follows, favoring the intervention: MD = 8.321 (95% CI = 3.255, 13.388). There was no heterogeneity, and the findings were significant (*p* = 0.0013, Z = 3.22, τ^2^ = 0, I^2^ = 0%).

#### 3.6.5. ALT

Three of the four studies reported ALT findings at the end-line (Figure 6); overall, the ALT outcomes were still high among those who underwent the intervention; hence, it is difficult to draw conclusions. The mean difference was as follows: MD = 40.306 (95% CI = 13.241, 67.37, *p* = 0.0035). There was moderate heterogeneity present (Z = 2.92, τ^2^ = 0, I^2^ = 41.1%).

#### 3.6.6. Total Bilirubin

All four studies reported bilirubin outcomes (Figure 7). The mean difference was MD = −0.083 (−0.935, 0.769). Thus, the intervention’s outcomes for total bilirubin did not tend toward any direction. The findings were insignificant, with no heterogeneity present (*p* = 0.849, τ^2^ = 0, I^2^ = 0%).

### 3.7. Risk of Bias Assessment Findings

For the randomized trials (*n* = 3), two studies expressed some concerns about biases arising from the randomization process and in the selection of the reported results, whereas no studies had concerns relating to biases arising from deviations from the intended interventions or to the measurement of the outcomes. One RCT had concerns about bias due to missing outcome data (Figure 8). Overall, all RCTs had a low risk of bias.

For the non-randomized clinical trial (*n* = 1), there were moderate concerns regarding bias due to confounding. There were low concerns about bias due to the selection of participants, classification of interventions, deviations from intended interventions, missing data, measurement of outcomes, and selection of the reported result. Overall, there was low risk in the study (Figure 8).

## 4. Discussion

This study presents a systematic review and meta-analysis of the use of two pharmacological agents, maralixibat and odevixibat, for Alagille syndrome in the treatment of cholestasis. A total of 94 patients across four studies with evidence of cholestasis as part of Alagille syndrome were evaluated. A significant reduction of −75.8 μmol/L was found for serum bile acid (sBA) as a treatment effect. Itch severity, graded with the Itch-Reported Outcome (ItchRo) score on a 0–4 point scale, was significantly reduced by 1.8 points. Two scales assessing the quality of life (QoL), the 100 point Multidimensional Fatigue Scale and the 100 point Pediatric Quality of Life scale, showed significant improvements of 11.4 and 8.3 points, respectively. Liver function tests did not show improvements; rather, alanine aminotransferase (ALT) levels were raised by 40 U/L and there was no significant difference in total bilirubin (TB) levels. This study evaluated phase 2 trials involving pediatric patients with AGLS and found a significant reduction from baseline for sBA and pruritus with different doses of maralixibat/odevixibat compared to placebo.

The Itch-Reported Outcome (ItchRO) is specifically developed for pruritus in AGLS and uses both patient- and observer-dependent reporting with scores ranging from 0 to 4, 4 being the most severe itching in one day observed by caregivers [26]. Daily ItchRO scores are documented in a diary and the overall score is calculated as an average depending on the screening period, which typically takes place over 2 weeks [26]. Pruritus and itching have a significant psycho-social impact on patients, which may be difficult to measure objectively [27]. In the case of AGLS, many patients are in early childhood, which makes self-reported scales, such as the visual analog scale [23], Itch Man Scale [28], and 5-D Itch Scale [29], difficult to use. The ItchRO scale has been piloted and validated with 12 AGLS patients, 24 caregivers, and 25 families, with a mean age of 8.3 years (0.44–34.9 years) for patients. There are two different measures in the ItchRO. The patient-rated measure is designed for patients aged 9 years or older, whereas the observer-related measure is designed for caregivers of younger individuals. Overall, the ItchRO is the most comprehensive measure of pruritus for cholestatic pruritus in AGLS [26]. However, there are certain discrepancies between pruritus and the severity of serum bile acids (and correlated biomarkers of pruritus), as the relationship between them is not linear, which has been demonstrated by Kamath and colleagues [30]. Actigraphy, which objectively measures sleep parameters and average motor activity noninvasively over a period of days to weeks, has been suggested, but it has not been validated with AGLS-associated pruritus [31]. Our meta-analytical findings suggested a significant reduction in ItchRO scores by 1.8 points, which lends support to the use of maralixibat/odevixibat for pruritus and the employment of ItchRO scores as surrogate endpoints in the clinical trials. We also found a parallel improvement in itch scores and reduction in sBAs, which suggests that IBAT may contribute to both subjective and objective improvement in AGLS patients.

Ileal bile acid transporter (IBAT), also known as apical sodium-dependent bile acid transporter (ASBT, *SLC10A2*), reabsorbs about 95% of the synthesized bile acids (BAs) when secreted into the small intestine. BAs are hydroxylated steroids made of excess cholesterol from the liver that is to be removed from the body, and they help with lipid digestion in the intestine. Only near 5% of BAs are excreted and the rest are reabsorbed in the terminal ileum and returned to the liver via the portal vein for use again (Figure 9). Cholestasis occurs due to reduced biliary flow secondary to impaired secretion by hepatocytes in AGLS [32]. Clinically, elevated serum levels of conjugated bilirubin or bile salts are commonly measured; pruritus (itch) is a common symptom present in cholestasis associated with ALGS that impacts quality of life significantly. Up until recently, patients with AGLS did not have any treatment options for refractory pruritus. As of now, maralixibat, an IBAT inhibitor, is considered a strong candidate by the Food and Drug Administration (FDA) as a Breakthrough Therapy and Orphan Drug for pruritus associated with AGLS among patients 1 year or older, building the roadmap for its approval by the FDA and gathering support from data from clinical trials for its strong efficacy [33,34]. Our findings support both maralixibat and odevixibat, both IBAT blockers, as important treatment options for AGLS, the use of which can lead to significant reductions in the severity of pruritus.

The benefits of reducing serum BAs in cholestasis associated with AGLS include the prevention of early liver fibrosis, inflammation, cirrhosis, and cancer [35]. BAs, when elevated, act as toxins that cause chronically inflamed states and cell death [36,37]. The underlying state of liver cells in cholestasis is marked by reduced regenerating capacity [38] and repeated injury to the cell architecture [36]. Elevated BAs may also act as tumor promoters and contributors to the development of hepatocellular carcinoma (HCC) [39]. IBAT inhibitors disrupt known mechanisms of reabsorption of BAs into the enterohepatic circulation, which may ameliorate the severity of cholestasis-associated morbidity in AGLS patients [40]. A significant sBA reduction of −75.8 μmol/L strongly suggests beneficial therapeutic effects from IBAT inhibitors in AGLS patients, which would be concomitantly supplemented by a reduction in pruritus, as noted earlier. However, treatment with IBAT inhibitors was associated with elevations in ALT levels, which may have been a treatment effect. It remains to be seen whether different IBAT inhibitor dosing regimens may improve sBAs and severity of pruritus without causing drug-induced hepatotoxicity [41] such that the benefits of IBAT inhibitors may be retained without compromising liver function integrity in AGLS patients.

### 4.1. Strengths and Limitations

Our study has many strengths. The four trials included in this systematic review and meta-analysis included the latest evidence relating to IBAT inhibitors in AGLS patients and were obtained through a robust search strategy and screening process. Two independent reviewers screened the initial search and the full texts with 88% agreement. As part of the systematic review, all systematic reviews and narrative reviews that reported AGLS-related outcomes were screened for potential studies. While none were included, our strategy ensured that no studies were omitted. Another strength was the double-checking of the data entry by a third reviewer. The conclusions drawn from this systematic review and meta-analysis were based on all the studies in the current literature that were of high quality. Another important strength was the nature of our results. The quality-of-life indicators, including the Multidimensional Fatigue Scale and the Pediatric Quality of Life scale, mimicked the itch severity and reduction in sBA. Ours is the first meta-analytical paper to report the efficacy of IBAT blockers in AGLS patients. We included all trials that reported the use of IBAT in AGLS patients. The efficacy of IBAT is typically measured as the reduction in pruritus and/or in sBAs, and our findings provide support for both in AGLS patients.

There are certain limitations to this study. The trials used slightly different criteria to include patients with AGLS, which may have represented different severities of pruritus. The dosing regimens varied across all trials and we could not ascertain the most efficacious dosing regimen for patients with AGLS. The duration of follow-up was also different across the trials, which may have led to under- or over-representation of the treatment effects of IBAT inhibitors in these patients. The outcome measures were variable; however, ItchRO scores and sBAs were reported across all trials, which are important surrogate clinical endpoints for AGLS patients. Odevixibat was given for a shorter duration than maralixibat across the trials. The ASSERT phase 3 trial [42] (NCT0467461) is currently evaluating the efficacy and safety of the use of odevixibat in patients with Alagille syndrome daily for 24 weeks. However, the itch measurement is different (an Albireo Observer-reported scratching score called PRUCISION [43]), which may make the results of efficacy comparison across clinical trials different. Discrepancies in endpoints were present in the current study and are expected in ongoing trials.

### 4.2. Recommendations

The most common adverse events reported across the trials included in this study were gastrointestinal, mainly diarrhea and abdominal pain, which was likely due to the mechanism of action of IBAT blockers. As IBAT blockers prevent BAs from re-entering the enterohepatic circulation, BAs are diverted to the large intestine, which may lead to such side effects. However, this may impact compliance with IBAT inhibitors. As effects are primarily gastrointestinal and may be dependent on dose, it may be necessary to consider optimizing the dosing regimen for long-term use in AGLS patients. For instance, lowering the dose or dosing interval of maralixibat or odevixibat in AGLS patients might provide relief from drug-related adverse effects without compromising efficacy. IBAT inhibitors reduce the levels of BAs in a short period of time, which is not immediately compensated for by liver synthesis of Bas [44]. For such calculated dosing, it may be pertinent to consider trials with longer dosing intervals and to compare surrogate endpoints (e.g., ItchRO and sBA) to optimize the treatment for AGLS. Another recommendation is to consider using similar endpoints to reduce the discrepancy in efficacy measures. However, there is certainly variability in the objective reporting of itching that cannot be accounted for [45]. Certain patients may be more sensitive than others and may self-report higher scores [46]. Identification of the best dosing regimen based on the subjective improvement in pruritus while maintaining liver function tests (LFTs) may be tailored to each patient.

## 5. Conclusions

In this systematic review and meta-analysis, 94 patients with Alagille syndrome from across four trials were included. Patients received either maralixibat or odevixibat in the intervention groups or a placebo. There was a significant improvement in Itch-Reported Outcome (ItchRO) scores by 1.8 points on a 0–4-point scale. This improvement was physiologically corroborated by a significant reduction in serum bile acid levels (sBAs) of −75.8 μmol/L, and all quality-of-life indicators in the trials were significantly improved. At the cost of elevated alanine aminotransferase (ALT) levels, ileal bile acid transporter (IBAT) inhibitors have favorable outcomes in patients with AGLS. Further trials may consider spacing the dosing frequency to reduce the gastrointestinal side effects while still retaining the positive effects on pruritus and sBAs.

## Figures and Tables

**Figure 1 jcm-11-07526-f001:**
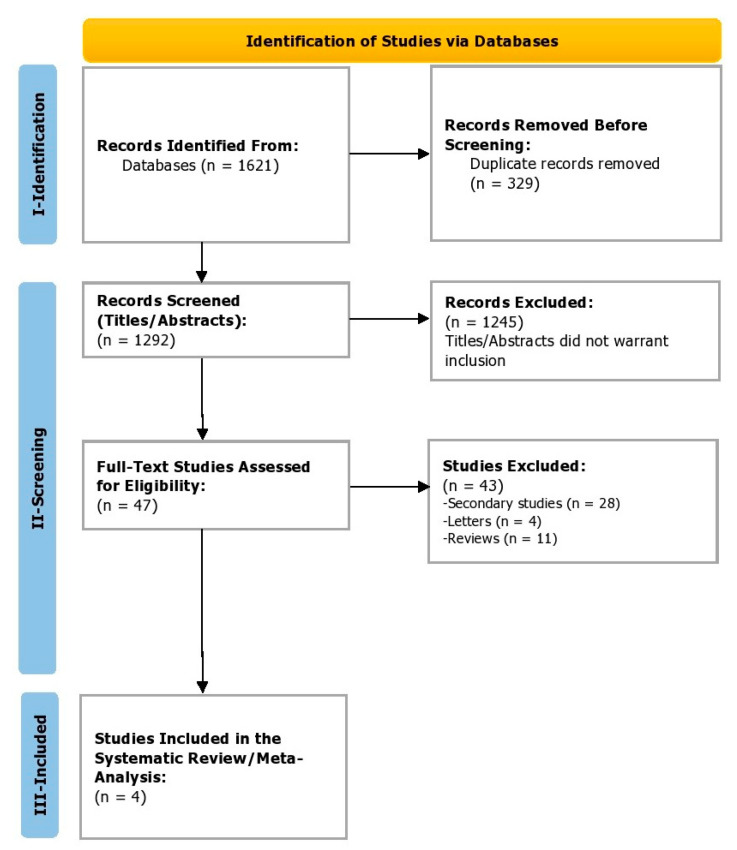
PRISMA study selection process.

**Figure 2 jcm-11-07526-f002:**
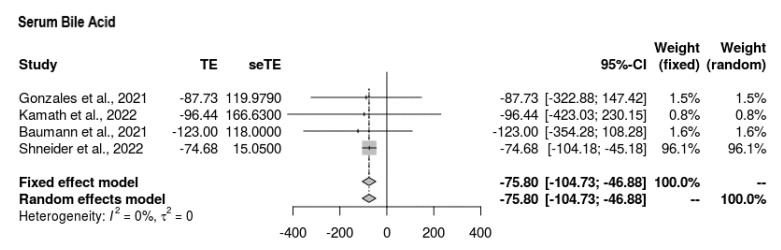
Forest plot depicting serum bile acid outcomes [19,20,21,22].

**Figure 3 jcm-11-07526-f003:**
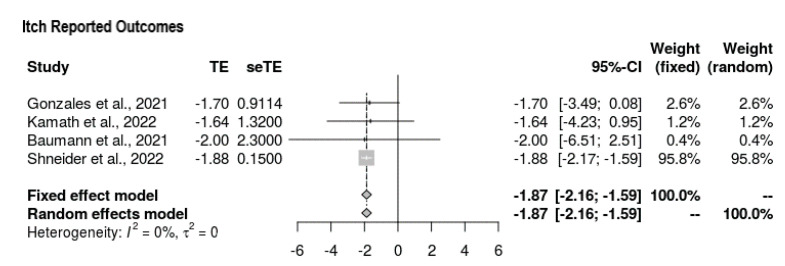
Forest plot depicting Itch Scale outcomes [19,20,21,22].

**Figure 4 jcm-11-07526-f004:**
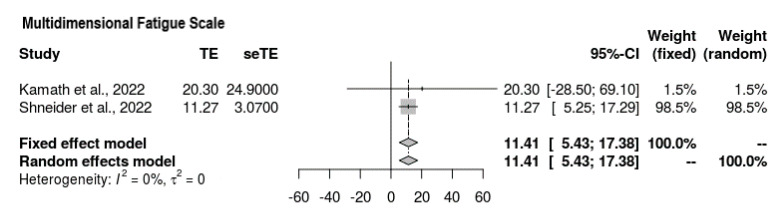
Forest plot depicting Multidimensional Fatigue Scale outcomes [21,22].

**Figure 5 jcm-11-07526-f005:**
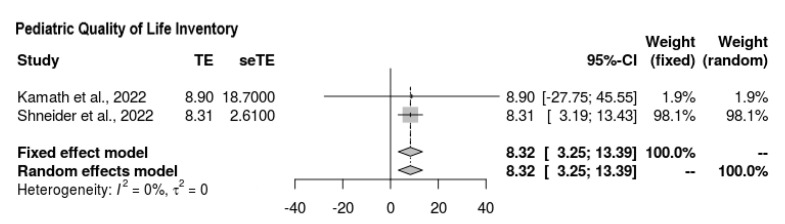
Forest plot depicting PedsQL score outcomes [21,22].

**Figure 6 jcm-11-07526-f006:**
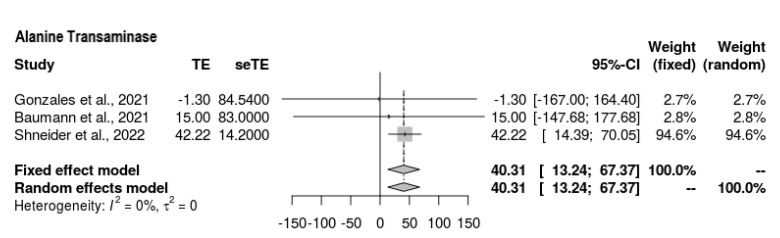
Forest plot representing ALT outcomes at end-line compared to baseline [19,20,21].

**Figure 7 jcm-11-07526-f007:**
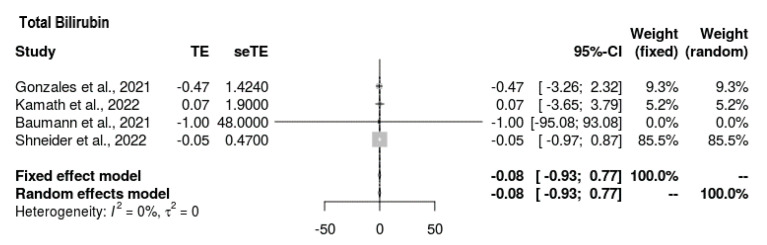
Forest plot depicting total bilirubin outcomes at end-line compared to baseline [19,20,21,22].

**Figure 8 jcm-11-07526-f008:**
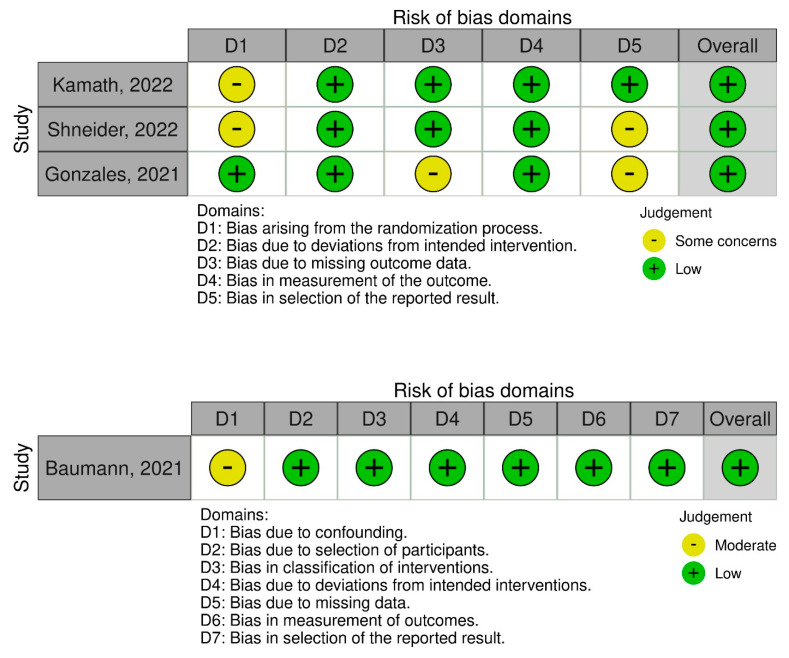
Risk of bias assessment of RCTs using the ROB 2 and ROBINS-I tools. The traffic light plots depict study-by-study bias assessment [19,20,21,22].

**Figure 9 jcm-11-07526-f009:**
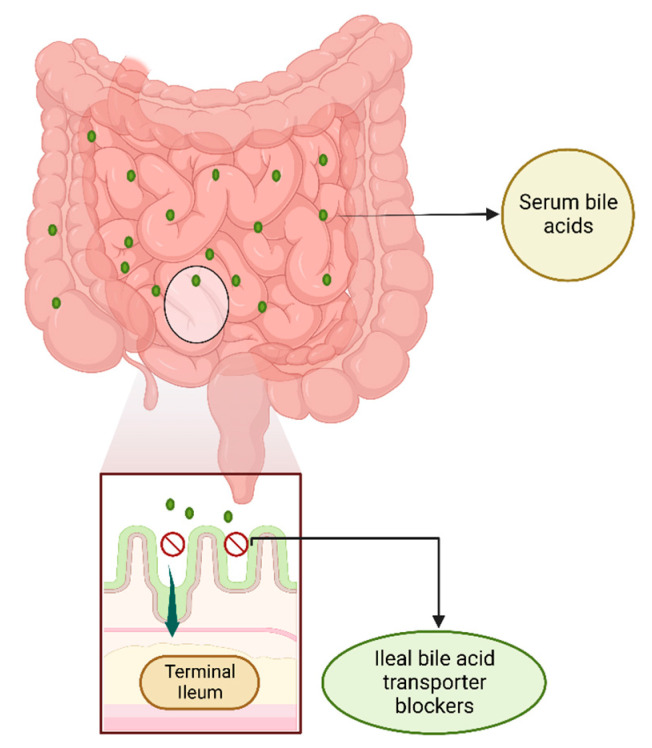
Ileal bile acid transporter (IBAT) blockers in the terminal ileum. Bile acids (BAs) are synthesized in the liver from cholesterol. Common Bas, such as cholic, deoxycholic, chenodeoxycholic, and lithocholic acid, may be combined with glycine or taurine to form conjugated BAs. Once excreted from the liver into the small intestine, most of the conjugated and unconjugated BAs are reabsorbed from the terminal ileum via the ileal bile acid transporter (IBAT) (or apical sodium-dependent bile acid transporter (*ASBT*, *SLC10A2*). The BAs reenter the portal circulation with the sodium-dependent taurocholate co-transporting peptide (NCTP, *SLC10A2*) via the portal vein (not shown). IBAT blockers disrupt the enterohepatic circulation by preventing the uptake of primarily conjugated Bas, whereas unconjugated BAs may be taken back in the liver through organic anion transporters that are less well-defined.

**Table 1 jcm-11-07526-t001:** Baseline characteristics of the included studies.

No.	Title	Author	Journal	Year	Phase	Design	Inclusion Criteria
**1**	Efficacy and safety of maralixibat treatment in patients with Alagille syndrome and cholestatic pruritus (ICONIC): A randomized phase 2 study	Gonzales	*The Lancet*	2021	Phase 2 trial (NCT02160782)	Placebo-controlled, randomized withdrawal period, phase 2b study with an open-label extension	Pediatric patients aged 1–18 years with Alagille syndrome
**2**	Maralixibat treatment response in Alagille syndrome is associated with improved health-related quality of life	Kamath	*The Journal of Pediatrics*	2022			
**3**	Effects of odevixibat on pruritus and bile acids in children with cholestatic liver disease: Phase 2 study	Baumann	*Clinics and Research in Hepatology and Gastroenterology*	2021	Phase 2 trial (NCT02630875)	Open-label, non-randomized, multicenter, single- and multiple-dose	Pediatric patients aged 1–18 years with pruritus due to chronic cholestatic disease (including Alagille syndrome), elevated serum total bile acids ≥2 times the upper limit of normal (ULN), and a score of ≥3 on an 11 point visual analog scale (VAS) for itch averaged over 7 days
**4**	Impact of long-term administration of maralixibat on children with cholestasis secondary to Alagille syndrome	Shneider	*Hepatology Communications*	2022	Phase 2 trials (NCT01903460, NCT02057692, NCT02047318 and NCT02117713)	Randomized, placebo-controlled, double-blind trial	Pediatric patients aged 2–18 years with Alagille syndrome, evidence of cholestasis, intractable pruritus, compensated liver disease, and a mean daily ItchRO (Obs) score of ≥2 for two consecutive weeks

**Table 2 jcm-11-07526-t002:** Baseline characteristics of the included studies.

No.	Author	Pharmacologic Agent and MOA	Intervention	Outcome Measures	Follow-Up
1	Gonzales	Maralixibat inhibits the apical sodium-dependent bile acid transporter	18 weeks of maralixibat 380 μg/kg once per day, followed by randomization (1:1) into groups that continued maralixibat or received a placebo for 4 weeks, then open-label maralixibat until week 48, followed by the long-term extension (up to 204 weeks and doses increased up to 380 μg/kg twice per day) *	(1) Change in mean serum bile acid (sBAs) during the randomized withdrawal period in participants with at least 50% sBA reduction by week 18, (2) cholestatic pruritus (0–4 point scale rated by the observer, patient, and clinician)	Baseline, 18 weeks, 22 weeks, 48 weeks
2	Kamath			(1) Itch-Reported Outcome (observer) score from baseline to week 48, (2) Pediatric Quality of Life Inventory Generic Core scores, (3) Family Impact scores, (4) Multidimensional Fatigue Scale scores	Baseline, 48 weeks
3	Baumann	Odevixibat, a potent, selective, reversible ileal bile acid transporter inhibitor	Single dose followed by a 14 day safety observation period; then, given daily for 4 weeks at the same dose as the initial single dose (10, 30, 60, 100, or 200 μg/kg)	(1) Change in serum bile acid levels, (2) VAS-itch (0–10 point scale), (3) Whitington itch (0–4 point scale), and (4) Partial Patient-Oriented Scoring Atopic Dermatitis (PO-SCORAD) itch and sleep disturbance score (0–10 point scale) (all scores self- or observer-reported daily and averaged over 7 days)	Baseline, 4 weeks
4	Shneider	Maralixibat inhibits the apical sodium-dependent bile acid transporter	Daily orally administered maralixibat (ranging from 140 to 560 μg/kg)	(1) Mean change in sBAs from baseline to weeks 48 and 72, (2) ItchRO (Obs) scores (0–4 point scale), (3) Clinician Scratch Scale (0–4 point scale) from baseline to weeks 48 and 72, (4) Quality of life: Parent PedsQL and Multidimensional Fatigue Scale (MFS) module (0–100 point scale with a higher score associated with a higher quality of life) from baseline to weeks 48 and 72, and (5) liver function tests from baseline to weeks 48 and 72	Baseline, 48 weeks, 72 weeks

* It should be noted that the studies by Gonzales and Kamath stemmed from the same primary clinical trial: “Safety and Efficacy Study of LUM001 (Maralixibat) with a Drug Withdrawal Period in Participants With Alagille Syndrome (ALGS) (ICONIC)” with NCT number NCT02160782.

**Table 3 jcm-11-07526-t003:** Patient characteristics, outcomes of the studies, and GRADE quality assessment scores.

Author, Year	*n*	Age (years)	Gender (% Male)	Efficacy	Safety	Remarks	GRADE Scores
Gonzales	31 patients	Mean age: 5.4 years (SD: 4.25)	19 (66%)	During the randomized withdrawal period (RWD), the least-square mean difference was −117 μmol/L (95% CI: −232 to −2); in the RWD, the placebo group had significant increases in sBA (94 μmol/L (95% CI: 23 to 164)) and pruritus (1·7 points (95% CI: 1·2 to 2·2)). From baseline to week 48, there were changes in sBA (−96 μmol/L, −162 to −31) and pruritus (−1·6 pts, −2·1 to −1·1)	Well-tolerated (mild-to-moderate events, mostly gastro-intestinal)	Improvements were seen in sBA, pruritus, and fatigue among children with Alagille syndrome for chronic cholestasis with maralixibat therapy	High-quality evidence
Kamath	27 patients			20 out of 27 patients (74%) had reductions of one point or more in the Itch-Reported score at week 48; there were Multidimensional Fatigue Scale score mean changes of +25.8 points in responders and −3.1 points in non-responders (*p* = 0.03); Family Impact scores increased by +16.9 compared to non-responders over 48 weeks, controlling for baseline Family Impact score (*p* = 0.05); non-significant changes were found for Pediatric Quality of Life Inventory Generic Core scores			High-quality evidence
Baumann	6 patients	* Mean age: 6.5 (SD: 4.6)	* 15 (62.5) *	Mean change in sBA across the entire cohort at week 4: −123 ± 118 umol/L (−394–15). Five of six patients with Alagille syndrome showed reductions in sBA; mean VAS-itch change: −6.1 to 0.4; PO-SCORAD itch change: −6.7 to −0.03; Whitington itch change: −1.6 to 0.8; PO-SCORAD sleep change: −5.5 to 0.7	Two out of six patients (33.3%) with Alagille syndrome (cohort receiving 200 μg/kg) had highly elevated ALT and AST at baseline and at the end of 4 weeks	Improvements were seen in sBA, pruritus, and sleep disturbance among children with Alagille syndrome with orally administered odevixibat	Moderate-quality evidence
Shneider	57 patients	Mean age: 6.5 years		Mean change in sBA from baseline to week 48: −79.88 umol/L (−114.57, −45.19); mean change in ItchRO (Obs) from baseline to week 48: −1.59 points (−1.81, −1.36); mean change in CSS from baseline to week 48: −1.36 points (−1.67, −1.05); mean change in PedsQL from baseline to week 48: +10.17 points (4.48, 15.86); multi-dimension fatigue score from baseline to week 48: +13.97 (7.85, 20.08)	9/57 patients (15.8%) had treatment-emergent adverse events, 6 of them (10.5%) due to elevated alanine aminotransferase (ALT) or total bilirubin (TB)	There was improved pruritus and quality of life with oral maralixibat among children with Alagille syndrome	High-quality evidence

* It should be noted that the studies by Gonzales and Kamath stemmed from the same primary clinical trial: “Safety and Efficacy Study of LUM001 (Maralixibat) With a Drug Withdrawal Period in Participants With Alagille Syndrome (ALGS) (ICONIC)” with NCT number NCT02160782.

## Data Availability

All data utilized for the purpose of this study are available publicly and online.

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
