# Peer review of "Ileal Bile Acid Transporter Blockers for Cholestatic Liver Disease in Pediatric Patients with Alagille Syndrome: A Systematic Review and Meta-Analysis"

_jcm, 2022, doi:10.3390/jcm11247526_

Round 1

Reviewer 1 Report

Main Comments:

(1) This manuscript deals with the efficacy of ileal bile acid transporter blockers for cholestatic liver disease in pediatric patients with Alagille Syndrome. It offers a systematic review and a meta-analysis. Although it is limited by the data available in the literature and the most efficacious dosage regimen could not be ascertained, it provides an overview on the current state of research as well as some recommendations.

(2) "embrytoxon" (line 39) -> the correct term is embryotoxon.

(3) "pruritis" (line 45, lines 48/49, and many other times throughout the text) -> the correct term is pruritus.

(4) Table 1: The relationship between study No. 1 and study No. 2 should be explained in a footnote, so that readers can understand the tenth column of the table ("Interventions") independently from the rest of the article.

Additional Comments/Suggestions:

(5) The page numbers (in the top right corner of the pages) are not correct; e.g., "2 out of 21" is used four times.

(6) Line 65: "adhered to adhering to…" -> adhered to…

(7) Line 84: "spreadsheet. the third reviewer…" -> spreadsheet. The third reviewer…

(8) Line 172: "Schneider et al." -> Shneider et al.

(9) Line 340-342: "Both different measures of ItchRO, including patient- and observer-rated, is designed for patients aged 9 years older and caregiver of those who are younger, respectively" – please improve this sentence.

(10) Lines 364/365: "Maralixibat, an IBAT" -> Maralixibat, an IBAT Inhibitor.

(11) Figure legend 9: "Ileal bile acid transporter blockers (IBAT)" -> Ileal bile acid transporter (IBAT) blockers.

(12) Lines 392-396: "However, IBAT treatment was associated with elevations in ALT levels which may be a treatment effect. It remains of interest whether different IBAT dosing regimens may improve the sBAs and pruritis severity without causing drug-induced hepatotoxicity [37] such that the benefits of IBAT may be retained without compromising liver function integrity in AGLS patients" -> However, treatment with IBAT inhibitors was associated with elevations in ALT levels which may be a treatment effect. It remains of interest whether different IBAT inhibitor dosing regimens may improve the sBAs and pruritus severity without causing drug-induced hepatotoxicity [37] such that the benefits of IBAT inhibitors may be retained without compromising liver function integrity in AGLS patients.

(13) Line 429: "ABAT blockers" -> IBAT blockers.

(14) Please improve Reference 30.

Author Response

Reviewer 1 Comments and Responses:

Main Comments:

Comment 1: This manuscript deals with the efficacy of ileal bile acid transporter blockers for cholestatic liver disease in pediatric patients with Alagille Syndrome. It offers a systematic review and a meta-analysis. Although it is limited by the data available in the literature and the most efficacious dosage regimen could not be ascertained, it provides an overview on the current state of research as well as some recommendations.

Response to reviewer comment: Thank you for your analysis. It is greatly appreciated.

Comment 2: "embrytoxon" (line 39) -> the correct term is embryotoxon.

Response to reviewer comment: Thank you for noting the discrepancy. It has been updated.

Comment 3: "pruritis" (line 45, lines 48/49, and many other times throughout the text) -> the correct term is pruritus.

Response to reviewer comment: Thank you for noting the discrepancy. It has been updated word by word throughout the manuscript. It is highlighted in yellow.

Comment 4: Table 1: The relationship between study No. 1 and study No. 2 should be explained in a footnote, so that readers can understand the tenth column of the table ("Interventions") independently from the rest of the article.

Response to reviewer comment: That is indeed a very helpful comment. Please note that I have made updates to the table’s footnote with the following: 

“*It must be noted that both studies by Gonzales and Kamath stem from a same primary clinical trial: “Safety and Efficacy Study of LUM001 (Mara-lixibat) With a Drug Withdrawal Period in Participants With Alagille Syndrome (ALGS) (ICONIC)” with a NCT number of (NCT02160782).”

Additional Comments/Suggestions:

Comment 5: The page numbers (in the top right corner of the pages) are not correct; e.g., "2 out of 21" is used four times.

Response to reviewer comment: All page numbers appear in their correct order now. Your are requested to have a look. Thank you very much for noting the discrepancy.

Comment 6: Line 65: "adhered to adhering to…" -> adhered to…

Response to reviewer comment: Thank you for noting the error. It has been fixed. Please review the change.

Comment 7: Line 84: "spreadsheet. the third reviewer…" -> spreadsheet. The third reviewer…

Response to reviewer comment: Thank you for noting the error. It has been fixed. Please review the change.

Comment 8: Line 172: "Schneider et al." -> Shneider et al.

Response to reviewer comment: Thank you for noting the error. It has been fixed. Please review the change.

Comment 9: Line 340-342: "Both different measures of ItchRO, including patient- and observer-rated, is designed for patients aged 9 years older and caregiver of those who are younger, respectively" – please improve this sentence.

Response to reviewer comment: This has been updated to the following:

“There are two different measures of ItchRO. The patient-rated one is designed for patients aged 9 years or older, whereas the observer-related one is designed for caregivers of younger individuals.”

Comment 10: Lines 364/365: "Maralixibat, an IBAT" -> Maralixibat, an IBAT Inhibitor.

Response to reviewer comment: Thank you for noting the error. It has been fixed. Please review the change.

Comment 11: Figure legend 9: "Ileal bile acid transporter blockers (IBAT)" -> Ileal bile acid transporter (IBAT) blockers.

Response to reviewer comment: Thank you for noting the error. It has been fixed. Please review the change.

Comment 12: Lines 392-396: "However, IBAT treatment was associated with elevations in ALT levels which may be a treatment effect. It remains of interest whether different IBAT dosing regimens may improve the sBAs and pruritis severity without causing drug-induced hepatotoxicity [37] such that the benefits of IBAT may be retained without compromising liver function integrity in AGLS patients" -> However, treatment with IBAT inhibitors was associated with elevations in ALT levels which may be a treatment effect. It remains of interest whether different IBAT inhibitor dosing regimens may improve the sBAs and pruritus severity without causing drug-induced hepatotoxicity [37] such that the benefits of IBAT inhibitors may be retained without compromising liver function integrity in AGLS patients.

Comment 13: Line 429: "ABAT blockers" -> IBAT blockers.

Response to reviewer comment: Thank you for noting the error. It has been fixed. Please review the change.

Comment 14: Please improve Reference 30.

Response to reviewer comment: The reference has been changed to a scholarly reference. Please review citations 33-34. Thank you for doing your due diligence.

--

I thank you for your time and dedication in helping us improve our work exponentially.

Best Regards,

Zouina!

Reviewer 2 Report

1. The authors were concerned with raising the maximum number of articles on the subject. However, why was it not included in the systematic search for the keyword “Diagnosis”? Could there have been more papers?

2. Line 48. “a wide array of patients undergo transplantation due to chronic pruritis." Inform percentage/number. Quantify.

3. Inform the mean age and sex of patients in the studies selected for analysis and by results. In some moments, 1 to 18 years appear, but not the average. This may be useful in future comparative studies.

4. Congratulations to use two tools for the risk of bias assessment. Excellent meta-analysis methodology.

5. The table can be formatted later with less spacing and more compact for viewing in the article.

Author Response

Reviewer 2 Comments and Responses:

Comment 1. The authors were concerned with raising the maximum number of articles on the subject. However, why was it not included in the systematic search for the keyword “Diagnosis”? Could there have been more papers?

Response to reviewer comment: We contemplated adding the keyword and added it to our keyword string. However, it limited our search so much that many articles were being omitted. We used an internal delphi methodology where we ran various keyword strings and stuck to the one with the most relevant studies (all 6 authors).

Comment 2. Line 48. “a wide array of patients undergo transplantation due to chronic pruritus." Inform percentage/number. Quantify.

Response to reviewer comment: I have scanned the entire internet and journal articles. This has been updated with the following: 

"A population-based study identifies that an estimated 7% of the general population may have chronic pruritus; while the global prevalence is unknown, a wide array of patients undergo transplantation due to chronic pruritus. However, a study reports that an estimated 15% of individuals with chronic pruritus require liver transplantation."

Comment 3. Inform the mean age and sex of patients in the studies selected for analysis and by results. In some moments, 1 to 18 years appear, but not the average. This may be useful in future comparative studies.

Response to reviewer comment: Your comment has been fully agreed with; I have updated the age in exact values as provided in the studies. Please review the text highlighted in yellow.

Comment 4. Congratulations to use two tools for the risk of bias assessment. Excellent meta-analysis methodology.

Response to reviewer comment: Thank you very much for your appreciation of our work. It is greatly appreciated and I thank you for giving your valuable time in reviewing it.

Comment 5. The table can be formatted later with less spacing and more compact for viewing in the article.

Response to reviewer comment: I broke down Table 1 into two separate tables. However, I believe once the paper is processed and ready for publication, the typesetting team really places it well. Please do not be concerned about that.

--

I thank you for your time and dedication in helping us improve our work exponentially.

Best Regards,

Zouina!